# Encounter rates and engagement times limit the transmission of conjugative plasmids

Jorge Rodriguez-Grande[1,2,3,4], Yelina Ortiz[1], Daniel Garcia-Lopez[1]
M. Pilar Garcillán-Barcia[1], Fernando de la Cruz[1], Raul Fernandez-Lopez[1]*

1 Instituto de Biomedicina y Biotecnologia de Cantabria IBBTEC, Spanish National Research Council CSIC – University of Cantabria, Santander, Spain, 2 Instituto de Investigación Sanitaria Valdecilla (IDIVAL), Santander, Spain, 3 Servicio de Microbiología, Hospital Universitario Marqués de Valdecilla, Santander, Spain, 4 CIBER de Enfermedades Infecciosas (CIBERINFEC), Instituto de Salud Carlos III, Madrid, Spain

* raul.fernandez@unican.es

## Abstract

Plasmid conjugation is a major route for the dissemination of antibiotic resistances and adaptive genes among bacterial populations. Obtaining precise conjugation rates is thus key to understanding how antibiotic resistances spread. Plasmid conjugation is typically modeled as a density-dependent process, where the formation of new transconjugants depends on the rate of encounters between donor and receptor cells. By analyzing conjugation dynamics at different cell concentrations, here we show that this assumption only holds at very low bacterial densities. At higher cell concentrations, conjugation becomes limited by the engagement time, the interval required between two successful matings. Plasmid conjugation therefore follows a Holling´s Type II functional response, characterized by the encounter rate and the engagement time, which represent, respectively, the density and frequency-dependent limits of plasmid transmission. Our results demonstrate that these parameters are characteristic of the transfer machinery, rather than the entire plasmid genome, and that they are robust to environmental and transcriptional perturbation. Precise parameterization of plasmid conjugation will contribute to better understanding the propagation dynamics of antimicrobial resistances.

## Author summary

Conjugative plasmids are semi-autonomous DNA molecules that carry antibiotic resistance genes and spread infectiously among bacteria. It is widely assumed that the rate of plasmid propagation depends on the density of susceptible recipient cells, as it is typical for infectious agents requiring contact for transmission. Our work reveals an additional, previously overlooked factor that constraints plasmid conjugation: the engagement time required for plasmid transmission. Using experimental and computational analyses, we show that plasmid conjugation dynamics follow a Holling´s Type II functional response, a class of ecological and infectious processes that are simultaneously limited by population density and the engagement time. We further show how these parameters depend on environmental and genetic factors, thus constraining the

**Data availability statement:** Data and code employed in the elaboration of all figures, tables and supplementary tables used in this manuscript is publicly available for analysis and reuse at https://github.com/IBBTEC/RodriguezGrande24

**Funding:** This work was supported by the Spanish Ministry of Science and Innovation Project PCI2021-122067-2A funded by MCIN/AEI /10.13039/501100011033 and the European Union Next Generation EU/PRTR and by Project PID2019-110216GB-I00 funded by MCIN/ AEI /10.13039/501100011033 to RF-L, and MCIN/AEI/10.13039/501100011033 PID2020-117923GB-I00 to FdlC and MPG-B, by the Spanish Ministry of Universities (FPU21/05415 to DG-L) and by the Spanish Ministry of Economy and Competitiveness (BES-2015-073463 to JR-G, linked to project BFU2014-55534-C2-1-P). The funders had no role in study design, data collection and analysis, decision to publish, or preparation of the manuscript.

**Competing interests:** The authors have declared that no competing interests exist.

speed at which a given plasmid may invade a susceptible population. Obtaining precise quantitative models for plasmid propagation is essential for curtailing the emergence of antibiotic-resistant infections.

## Introduction

In bacteria, a sustained encroachment of antimicrobial resistances is threatening our ability to treat infection. Antibiotic therapy is a safe and effective tool to fight bacterial pathogens, but its efficacy declines as the prevalence of antibiotic resistance genes (ARGs) increases. Bacterial conjugation is a key driving force in the spread of ARGs, which are frequently encoded in plasmids [1,2]. Conjugative plasmids transmit themselves from one cell to the other, in a process that resembles the spread of pathogens among susceptible populations [3–8]. Obtaining quantitative models for plasmid infection dynamics is key, not only for microbiologists studying the mechanisms of conjugation, but also for epidemiologists analyzing the spread of antibiotic resistances.

Since plasmid transmission requires cell-to-cell contact between a donor (D) and a recipient cell (R), conjugation has been classically assumed to follow mass-action-kinetics [4,9–11]. Seminal models posed that the rate of transconjugant (T) formation may be approximated as the product of a particular conjugation rate, $\gamma$, multiplied by the concentrations of [D] and [R] [12]. Such approximation is formally and conceptually identical to density-dependent models (DDT) for the transmission of infectious agents (Fig 1A). But the dynamics of plasmid conjugation are more complex than that of classical pathogens. In bacteria, the timescales of plasmid transmission and vegetative growth are of similar magnitude, thus new T cells may arise either by conjugation or by host replication [13,14]. Early models for plasmid conjugation, such as Simonsen´s End-Point Method, devised clever methods to deconvolve both contributions [3]. More modern approaches included further complexities of plasmid propagation, such as the effect of population spatial structure, or the stochastic nature of plasmid conjugation [15–18]. Although the quantitative framework to study conjugation gained in complexity along the years, most models kept as axiomatic that the conjugation rate shall be proportional to D and R concentrations [10,19–21]. While this can be logically drawn from first principles (conjugation cannot happen without physical contact), its validity along the natural range of bacterial densities cannot be taken from granted. It may be possible that the rate of encounters ceases to be a limiting factor as cells reach a certain concentration. This kind of density limitation occurs in the transmission of some infectious agents [22,23]. Many sexually transmitted infections do not follow DDT dynamics but instead can be better modeled using frequency-dependent transmission (FDT, Fig 1B). In ecology, a similar situation occurs in predator-prey dynamics. Predation rates are limited, not only by the availability of prey, but also by the time required for the predator to consume its catch [24]. This double dependence generates a biphasic regime, in which DDT becomes dominant at low prey densities, while FDT dominates at higher ones (Fig 1C). These dynamics can be modeled using Holling´s type II functional response, which smoothly captures this biphasic regime using two parameters: the encounter rate ($K_{on}$, which governs the DDT regime), and the engagement time ($\tau$, which sets the FDT limit).

Here, by studying the dependence of plasmid transmission rates on recipient densities, we found that conjugation follows a Holling´s type II functional response. Plasmid transmission at a wide range of bacterial concentrations can be faithfully modeled from its $K_{on}$ and $\tau$ values. These values were found to vary in different model broad host range plasmids, and

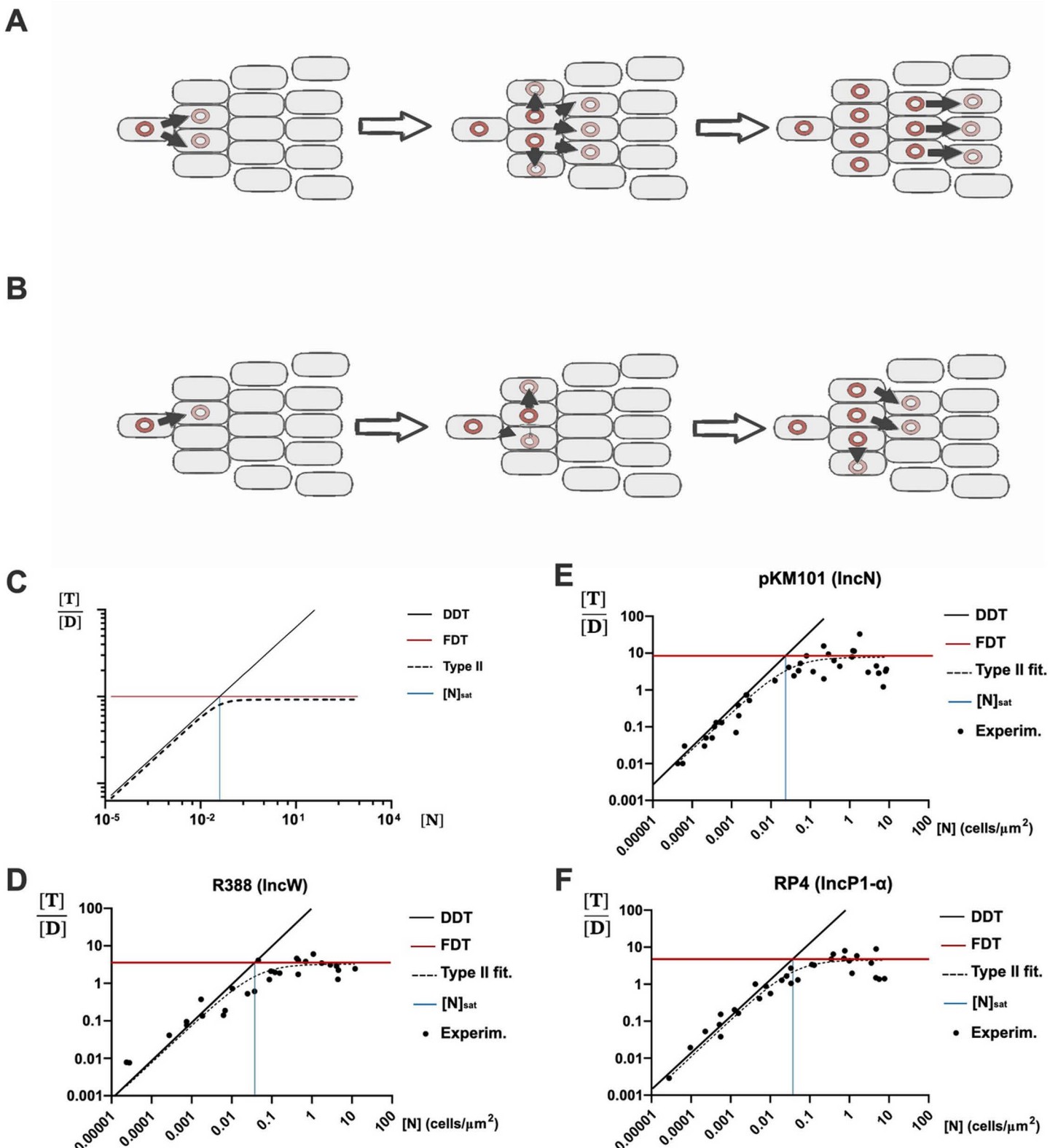

**Fig 1. Conjugation follows a type II Functional response. A)** Scheme showing the conjugative transmission of a plasmid (red circle) in a population of susceptible cells under Density-Dependent Transmission (DDT). In this regime, conjugation (black arrows) occurs when a plasmid containing cell contacts a plasmid-free one. The number of transconjugant cells formed is thus equivalent to the number of Donor:Recipient encounters. The conjugation frequency thus increases linearly with

increasingly available recipients. **B)** Scheme showing the conjugative transmission of a plasmid (red circle) in the same population of susceptible cells as in A), but under Frequency-Dependent Transmission (FDT). In this regime, conjugation (black arrows) is limited by the time required to complete the transfer event, thus not all Donor:Recipient contacts result in successful Transconjugant formation. The conjugation frequency is thus constant, regardless of the number of available recipients. **C)** Ideal number of transconjugants obtained per donor (y axis, T/D) as a function of the total cell concentration (x axis, N). In a Holling´s Type II functional response there is a density-dependent region where transconjugant formation increases proportionally with N (leftwards from the blue line). When N becomes saturating the formation of transconjugants does not scale any further with cell density and becomes frequency-limited (rightwards from the blue line). **D, E, F)** Transconjugants per donor ([T]/[D], y axis) obtained at different cell densities ([N], x axis) in conjugations using E. coli BW27783, on solid LB-agar surfaces matings and 1 h. conjugation time. Each graph corresponds to plasmids R388 (D), pKM101 (E) and RP4 (F). The black line represents the ideal DDT dynamics, while the red line represents the FDT.

even among members of the same plasmid taxonomic unit (PTU). Higher encounter rates and lower engagement times correlated with plasmid that displayed faster invasion dynamics. Overall, our results demonstrate that conjugation rates are intrinsically limited at a wide range of bacterial densities, but these limitations vary from plasmid to plasmid, suggesting that they may be genetically encoded and under active selection. Quantifying the transmission dynamics of different plasmids may shed light on the reasons for the different prevalence of PTUs in the microbiota, and their role in the spread of ARGs.

## Results

### Plasmid transmission becomes frequency-limited at moderate cellular densities

In a pure DDT regime, the number of transconjugants formed by each donor cell increases linearly with cell density (Fig 1C). In a pure FDT, however, the number of T/D cells generated by conjugation remains constant regardless of D and R concentrations [25]. To analyse the transmission regime for conjugative plasmids we plotted the T/D ratio exhibited by three model broad-host range plasmids which are stable in *E. coli* and show high conjugation frequencies: pKM101 (IncN/PTU-N1), R388 (IncW/PTU-W) and RP4 (IncP/PTU-P1). Conjugation experiments were performed between Nalidixic (Nx) and Rifampicin (Rif) resistant isogenic *E. coli* strains on LB-agar plates, as described in Materials and Methods. To minimize the confounding effects of vegetative growth, short mating times (1h unless stated otherwise) were employed, and the overall growth of D and R+T cells in these conditions was measured to be negligible (S1 Fig). To counteract the possible effects of structured populations, 1:100 and 1:10000 D:R ratios were used, so to ensure that each D cell is surrounded by suitable recipients. In these conditions, results, shown in Fig 1E–F, revealed that the number of transconjugants produced per donor cell scaled linearly with overall cell density, as expected for a DDT process, but reached a maximum at a density of approximately 0.05 cells/µm². This transition density, named $[N]_{sat}$ for saturation density, roughly corresponds to 0.01 to 0.1 cells per square micron [3,10,19,26,27]. From this density onward, transmission followed an FDT dynamics, where conjugation rates were independent of the cellular concentration. Although the three plasmids changed from DDT to FDT regimes at approximately the same density, the transmission rate was different. Plasmid pKM101 showed a maximum rate of nearly 10 T cells per D per hour, higher than that of plasmid RP4 (~4 T/D*h) and plasmid R388 (~3 T/D*h).

To evaluate whether this transition from DDT to FDT dynamics was characteristic of the rigid pili used to conjugate by the plasmids above, we repeated the experiment using plasmids pOX38 and R64 (IncFI/PTU-F1 and IncI1/PTU-I1, respectively). These plasmids use a flexible, long pilus to conjugate at high frequencies in liquid media [28]. As shown in S2 Fig, plasmids showed a similar behaviour as pKM101, RP4 and R388, demonstrating that this transition is common among MPF families when conjugation was performed on a solid surface. To test whether this behaviour was also observed when conjugation was performed in liquid media, we repeated analysed plasmid R64 transmission in LB broth. As shown in S3 Fig,

T/D ratios saturated at cell densities >$10^6$ cells/ml, indicating that this DDT/ FDT transition also occurs in liquid media.

## Conjugation dynamics follows a type II functional response

Results from Fig 1. demonstrated that plasmid transmission did not follow pure DDT or FDT dynamics. To obtain an analytic model able to capture the entire dependence on cell density, we posed the following reaction scheme, where D, R and T correspond, respectively, to Donor, Recipient and Transconjugant cells.

$$D + R \underset{k_{off}}{\overset{k_{on}}{\rightleftarrows}} C \overset{kc}{\rightarrow} D + T \qquad \text{Eq.1}$$

In Eq.1, D and R cells encounter each following mass action kinetics with a $K_{on}$ rate. Formation of the conjugative pair, C, then ensues. C may resolve into the irreversible formation of a new transconjugant T at rate $K_c$, or, alternatively, in an infructuous mating with a reverse rate $K_{off}$. The engagement time, hereafter $\tau$, is the average time required for a donor to turn a receptor into a transconjugant, and can be expressed as $1/K_c$. The model described in Eq.1 is suited for the transfer of mobilizable plasmids, where transconjugants are not able to become donors themselves. However, minor modifications allow the model to be adapted to self-transmissible conjugative plasmids (S1 Calculations). Eq.1 reaction scheme is conceptually identical to a Michaelis-Menten dynamics, and the same approximations can be used to obtain an analytic expression for transconjugant formation. As previously shown [29], following Haldane´s approximation the number of transconjugants per donor (the conjugation frequency) obtained after time $t$ follows:

$$\frac{[T]}{[D]} = \frac{[R] \cdot k_{on}}{1 + [R] \cdot k_{on} \tau} t \qquad \text{Eq.2}$$

Eq.2 corresponds to Holling´s type II functional response, originally proposed as the relationship between predation rate and prey density [24]. In the context of plasmid conjugation, the density of susceptible recipient cells (R) divides the response into two distinct regimes (Fig 1C). At low R densities, the encounter rate ($k_{on}$) becomes limiting (density-dependent transmission). At high R densities, however, it is the engagement time, $\tau$, which limits plasmid propagation (frequency-dependent transmission). Holling´s type II functional response describes both regimes and the transition between them as the R density changes. Eq.2 is valid for plasmid mobilization, but as shown in S1 Calculations, plasmid conjugation can be similarly modelled. In this latter case, the autocatalytic nature of transmission (T cells become donors in conjugation, but not in mobilization) results into an exponential relationship of the form:

$$ln\left(1 + \frac{[T]}{[D]}\right) = \frac{[R] \cdot k_{on}}{1 + [R] \cdot k_{on} \tau} t \qquad \text{Eq.3}$$

To check the validity of our approximation, a stochastic model using Chemical Master Equations (CMES) was built (S1 Calculations). As shown in S4 Fig, Eq.3 was able to capture the conjugation frequency with 10% estimation error, when short conjugation times (1 generation time) were used.

## Differences in conjugation parameters result in faster transmission dynamics

Data shown in Fig 1 was fitted to Eq.3 as described in Materials and Methods, to extract the searching rate and engagement times of plasmids RP4, pKM101 and R388. Results are shown in Table 1. Results indicated that pKM101 had a higher encounter rate $K_{on}$ (*i.e.*, donors carrying this plasmid found more available recipients during the same mating interval) and a shorter engagement time, τ (*i.e.*, donors carrying this plasmid apparently need shorter times to successfully transfer the plasmid after contact), with respect to RP4 and R388.

To analyze the impact of differential encounter rates and engagement times on the overall kinetics of plasmid invasion, we monitored the progression of the plasmid in a susceptible population along time. Since plasmids pKM101 and R388 had previously shown the highest differences in engagement times and searching rates, we performed invasion assays with these two model plasmids in *E. coli* populations. As indicated in Materials and Methods, plasmid progression was measured as the number of transconjugant cells produced per donor cell (T/D). We performed these assays at two different D: R proportions, 1:100 and 1:10000, keeping the same total cell concentrations (*i.e.*, using less donors). Results, shown in Fig 2A, demonstrate that both plasmids were able to successfully invade the population after 180 min., reaching T/D>10 in both cases. Plasmid pKM101 was significantly faster at both D:R ratios tested, thus demonstrating that higher searching rates and lower engagement times result in faster invasion rates.

## Differences in the conjugation parameters are intrinsic to the conjugation machinery

Plasmid conjugation is a complex mechanism in which factors other than the transfer machinery often have an impact on the overall rate of plasmid propagation [30–32]. To test whether differences in encounter rates ($K_{on}$) and engagement times (τ) were intrinsic to the conjugation machinery, or depended on the entire plasmid genomes, we constructed a series of *E. coli* mutants harboring chromosomal constructions of the entire PTU-N1 or the PTU-W transfer systems respectively. For this purpose, the entire *tra* operon of each of the plasmids was introduced into the chromosome of *E. coli*. These $tra_N$ and $tra_W$ constructions allow the mobilization of plasmids encoding the cognate *mob* regions ($mob_N$ and $mob_W$). *Tra* genes from R388 or pKM101 were cloned into *E. coli* MDS42 and MG1655 chromosomes respectively under their own, native promoters as indicated in Materials and Methods. We first tested whether these constructions were able to mobilize their cognate *mob* regions (Fig 2B). We then compared the mobilization dynamics of *tra + mob* constructions with their cognate plasmids (Fig 2C and 2D). The strains harboring the *tra + mob* constructions achieved lower T/D ratios than their parental plasmids. This is not surprising, since secondary transfer from the transconjugants is absent in experiments with the *tra + mob* constructions (only the *mob* containing plasmid mobilizes to the recipient). To test whether the differences observed were solely due to the kinetic difference between conjugation and mobilization, we repeated R388 invasion

**Table 1. Conjugation parameters obtained by fitting Eq.3 to data shown in Fig 1.**

| Plasmid (PTU) | $K_{on}$ (Conf. interval at 95%) | τ (C.I. at 95%) |
|---|---|---|
| R388 (PTU-W) | 90 $\mu m^2 \cdot h^{-1}$ (45–130) | 0.33 h (0.25–0.4) |
| RP4 (PTU-P1) | 160 $\mu m^2 \cdot h^{-1}$ (115–200) | 0.25 h (0.2–0.3) |
| pKM101 (PTU-N1) | 215 $\mu m^2 \cdot h^{-1}$ (180–250) | 0.13 h (0.1–0.2) |

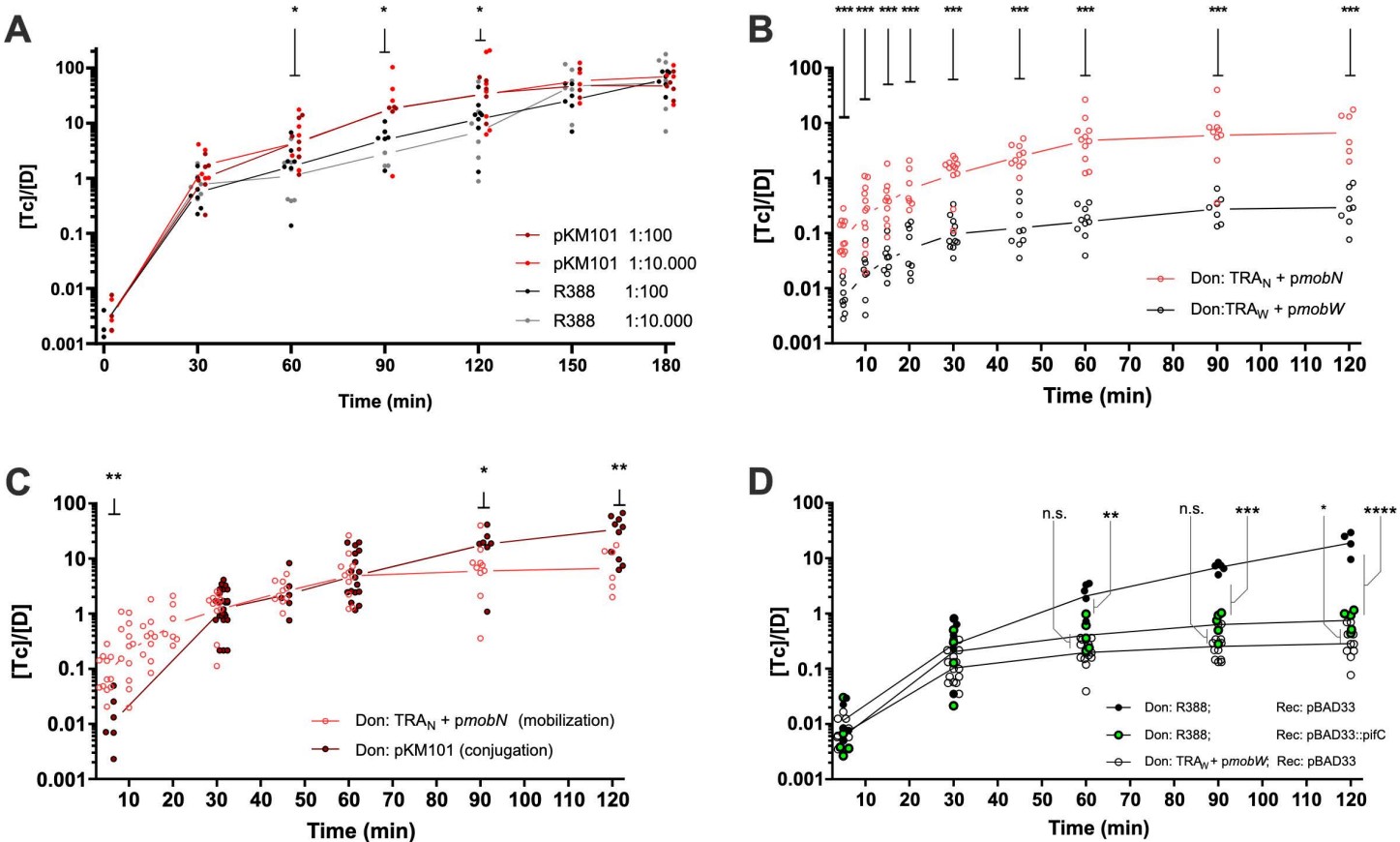

**Fig 2. Plasmid invasion assays.** Plasmid invasion of a susceptible population along time (x axis), expressed as transconjugants per donor (Y axis) **A)** Invasion dynamics of wild-type plasmids pKM101 (red dots) and R388 (black dots) obtained at two different donor: recipient ratios (1:100 and 1:10,000). Asterisks over t = 60, t = 90 and t = 120 minutes indicate statistically significant differences between plasmids pKM101 and R388 with $p < 0.05$ (*) B) Mobilization dynamics obtained with TRA genes from PKM101 ($tra_N$, red) and $tra$ genes from R388 ($tra_W$, black) integrated in the chromosome of E. coli. The mobilizable plasmid contained the MOB region of the corresponding plasmid. Differences between plasmids were statistically significant with $p < 0.01$(***) for all time points analysed C) Differences in the dynamics of conjugation (black) and mobilization (red) in plasmid pKM101. Mobilization assays were perform using a vector containing $mob_N$ harboured by a strain that contained the $tra_N$ genes inserted in the chromosome. Conjugation dynamics were obtained using the wild type pKM101 plasmid. Statistical differences were observed at t = 90 (* $p < 0.05$) and t = 120 (** $p < 0.01$) D) Comparison of plasmid R388 conjugation dynamics (black dots) vs R388-derived constructions. Black dots correspond to the wild-type plasmid transferred to a recipient strain that contains an empty pBAD expression vector. Green dots represent values obtained in conjugations where donors contained wild-type R388, and recipients harboured a pBAD::pifC construction, able to block secondary transmission. White dots correspond to the mobilization of a plasmid containing the $mob$ region of plasmid R388 from a strain harbouring a chromosomal integration of the Tra$_W$ genes. Black lines indicate comparisons between groups (* $p < 0.05$, ** $p < 0.01$, *** $p < 0.001$, **** $p < 0.0001$). When not explicitly indicated, the comparison between measurements was non statistically significant (n.s.).

assays in a population of recipient cells expressing PifC. PifC protein is an inhibitor of R388 conjugation [33], thus its presence in the receptors prevents transconjugants to become effective donors. Conjugation of plasmid R388 to a PifC-containing receptor should yield similar dynamics to mobilization from $tra_W$ if the difference between conjugation and mobilization is solely due to the absence of secondary transfer. As shown in Fig 2D, the transfer dynamics in $tra_W$ were similar to the conjugation of plasmid R388 to a population expressing PifC. This indicated that the differences between R388 and the strain harboring $tra_W$ can be mostly ascribed to the lack of secondary conjugation in the latter.

Once the functionality of the chromosomal constructions had been established, we compared the mobilization rates of $tra_W$ and $tra_N$. Results, shown in Fig 2B demonstrated that

the transfer machinery of pKM101 exhibited higher mobilization rates than R388. These results are consistent with the values obtained for wild type plasmids (Fig 2A) and cannot be ascribed to a faster growth of pKM101-containing cells (S5 Fig). Altogether, these results indicate that encounter rates and engagement times observed in full plasmids are dependent on the transfer apparatus of both plasmids.

## Differences in conjugation rates are robust to environmental variables and overexpression of the transfer machinery

Plasmids pKM101 and R388 showed significant differences in their encounter rates and engagement times, and these differences could be ascribed to their transfer machinery. However, environmental factors such as temperature, pH and nutrient availability affect the transfer efficiency of many plasmids [34]. Thus, it was possible that the differences observed between pKM101 and R388 could be explained by a differential sensitivity to environmental conditions. To test this end, we performed conjugations in different temperatures and nutrient compositions. Conjugation assays were performed as detailed in Materials and Methods, and results are shown in Fig 3A and 3B. As shown in Fig 3A, both plasmids exhibited an optimal transmission at 37 ºC, yet pKM101 showed higher transfer rates at all temperatures between 30 and 41 ºC. Plasmid conjugation rates were also relatively invariant when different media were used (Fig 3B): LB buffering against acidification (produced by carbon consumption) did not change the conjugation rates, as neither it did when we increased the nutrient availability by performing conjugations in terrific broth agar (TB agar). In all conditions tested, plasmid pKM101 consistently yielded higher conjugation efficiencies than R388. Also, no substantial differences among mating conditions were observed when using the $tra_W + mob_W$ mobilization setup. Altogether, results demonstrated that the differences between the rates of pKM101 and R388 were not caused by a differential sensitivity to environmental parameters.

Transcriptional regulation is another key factor modulating plasmid conjugation efficiency. Another possible explanation for the differences in the encounter rates and engagement times of both plasmids was a differential expression of the conjugation apparatus. The expression of conjugation genes in both plasmids is regulated through two conserved negative feedback loops. TRA operon is auto repressed by KorA, its first ORF, while MOB functions are controlled by second negative feedback under control of the accessory protein ($TrwA_{R388}$/$TraK_{pKM101}$). Changes in these repressors or their cognate operators could result in different expression levels, thus we tested whether increasing the expression of conjugation genes altered plasmid transmissibility. Specifically, we analyzed whether increasing the expression of transfer genes increased the conjugation frequency of R388. For this purpose, *mob* and *tra* operons, as well as each individual conjugation gene were cloned under the control of pBAD promoter and introduced into donor cells. Overexpression of these genes was achieved as indicated in Materials and Methods, and the conjugation rates determined as previously described [35]. Results, shown in Fig 3C revealed that the overexpression of transfer genes did not increase the conjugation frequency. Minor, yet statistically significative changes were achieved when overexpressing TrwJ, the whole *trwHGFED* operon or the truncated *trwMKJI* operon. TrwJ is the pilus adhesin, the protein responsible for contacting the recipient cell [36]. Thus, it is possible that overexpression of this protein results in better donor to recipient adhesion, [37–43], increasing the apparent $K_{on}$. The overexpression of the auto-repressed *korA-trwLMKJI* operon led to decreased conjugation rates, probably due to overexpression of the *korA* repressor acting *in trans* on R388 itself [44].

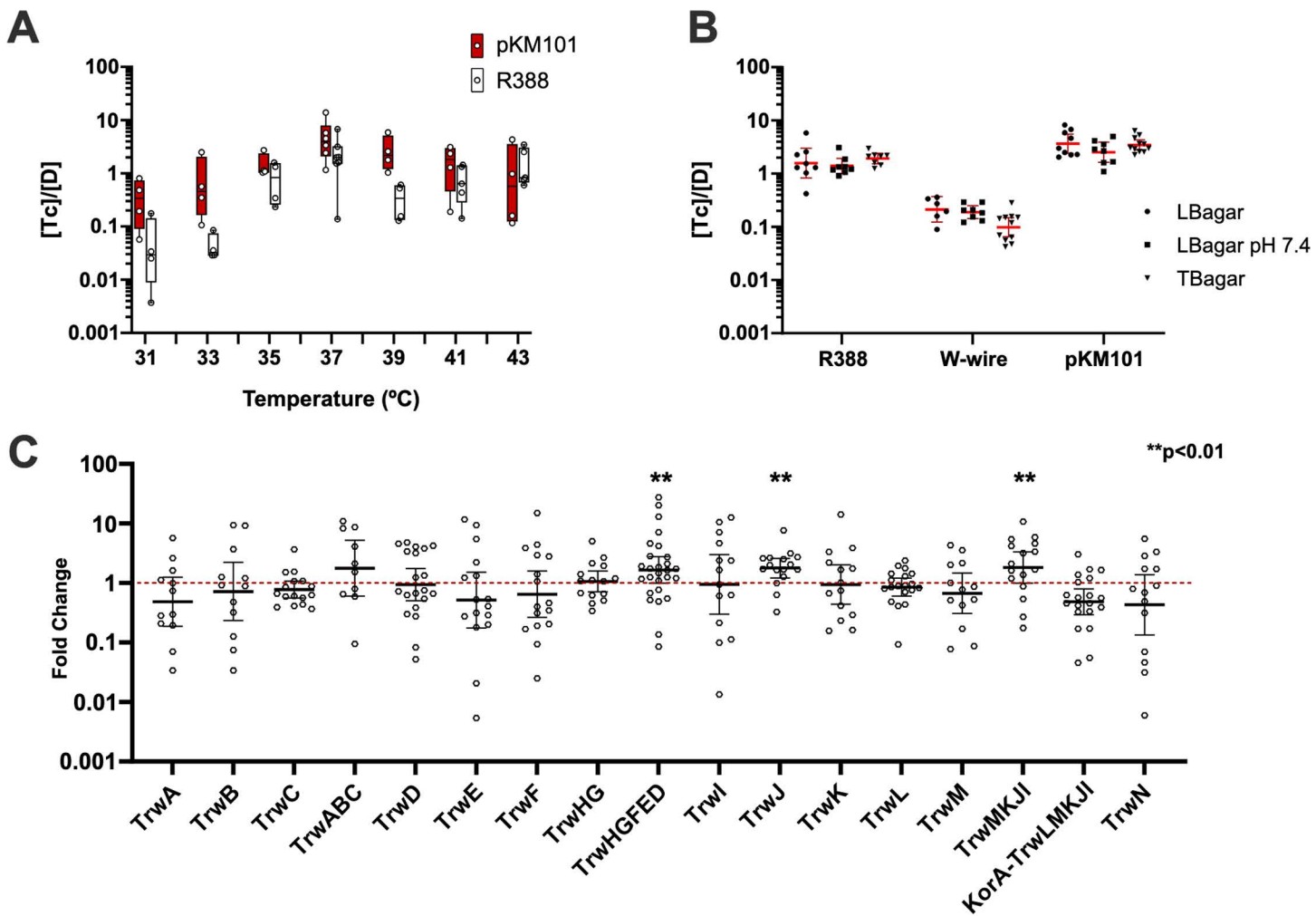

**Fig 3. Effect of environmental conditions and protein expression on conjugation rates. a) Effect of temperature on the conjugation rate.** Conjugation assays were performed as described in Materials and Methods. (1 h, 37 ºC, 1D:100R). Each point corresponds to an independent experiment **b)** Effect of media composition on the transfer rate for R388 conjugation and mobilization (W-wire) and pKM101 conjugation (LBagar: autoclaved LBagar, LBagar pH7.4: PBS-buffered at pH 7.4 and TB agar: Terrific Broth agar. **c)** Overexpression of *tra* genes does not increase the transfer rate. On the x axis, genes overexpressed in R388-carrying donors using a pBAD33 vector. On the y axis, the fold change between the conjugation frequency (as [T]/[D]) obtained overproducing each of the constructions shown in the figure, and the conjugation frequency obtained with an empty pBAD33 vector. Statistical significance was determined using Welch's corrected ANOVA test for samples with unequal variances.

## Variation in the fundamental parameters within the PTU-W

Results demonstrated that pKM101 and R388 exhibited different conjugation parameters, which resulted in different invasion dynamics. These differences could be ascribed to their transfer machineries and were considerably orthogonal to environmental and transcriptional perturbations. We thus wondered whether the encounter rate and the engagement time were characteristic of a given plasmid PTU, or were variable, with each plasmid showing different values. To test this end, we compared plasmid R388 with three other PTU-W plasmids: pMBUI4, pIE321 and R7K. PTU-W plasmids have a highly conserved genomic backbone [45], pIE321 being 97% identical at the DNA sequence level to R388, while R7K being 97.5% (S6 Fig). Despite this high level of sequence conservation, these plasmids were isolated from different bacterial species. Plasmid pIE321 was isolated from *Salmonella enterica* serotype Dublin

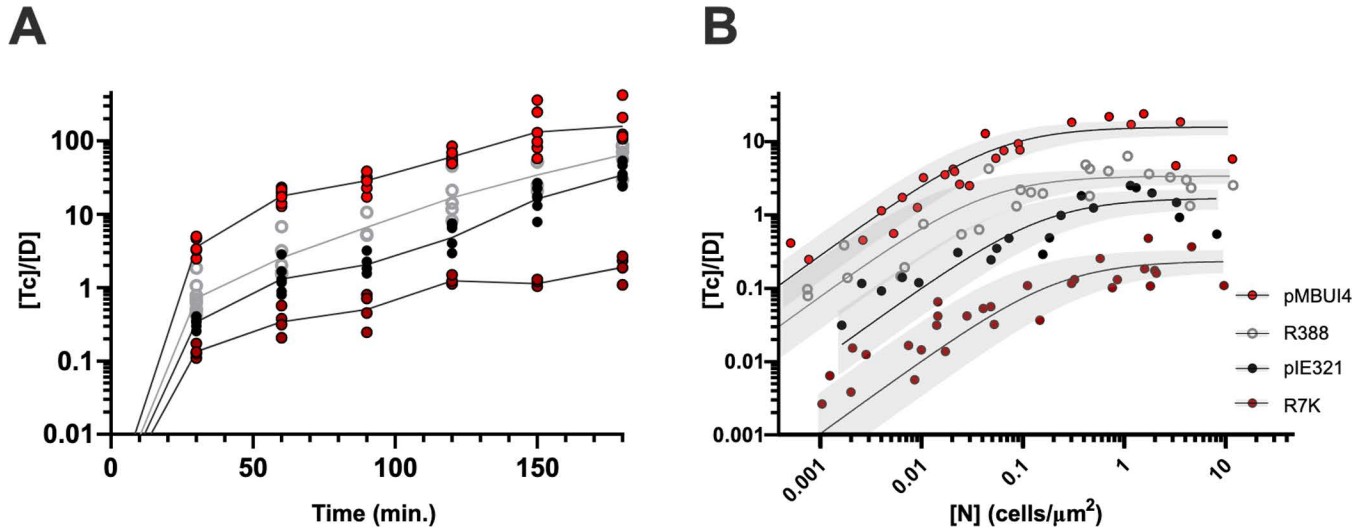

**Fig 4. Plasmids from the same PTU display different conjugation dynamics. (A)** Conjugation kinetics for different plasmids of the same PTU. The conjugation frequency, expressed as Tc/ D is indicated on the y axis. The conjugation time, in minutes, is indicated on the x axis. **(B)** Circles indicating conjugation frequencies (y axis) obtained at different cell densities (x axis) for different plasmids belonging to PTU-W. Black lines indicate the interpolated [T]/[D] values according to Eq.3. Grey areas indicate the predicted confidence interval at 95% for the adjusted $K_{on}$ and $\tau$ values. The estimated engagement times ($\tau$) and encounter rates ($k_{on}$), with their respective 95% confidence intervals are as follows: pMBUI4 Kon = 274 [131–580] µm2, $\tau$ = 4 [3.2–5.1]´; R388 Kon = 76 [28–263] µm2, $\tau$ = 18 [15–24]´; pIE321 Kon = 10 [4–32] µm2, $\tau$ = 36 [28–51]´; R7K Kon = 1 [0.3–4] µm2, $\tau$ = 262 [185–364]´.

[46], while R7K was found in *Providencia rettgeri* [47] and pMBUI4 directly from freshwater samples [48]. Searching rates and engagement times were obtained, as previously described, by analyzing the T/D obtained after 1h conjugation against a population of recipients with variable cell density (Fig 4B). While in pIE321 the engagement time was slightly higher than that of R388, we observed a 10x increase in the case of plasmid R7K. To check whether these differences had an impact on the speed of propagation of R7K, we performed a conjugation kinetics assay, as described before (Fig 4A). As shown in the figure, plasmid R7K invasion of the recipient population was much slower than that of R388, taking >150 minutes to reach a conjugation frequency of 1 T per donor cell, while R388 reached these levels in <30 min. Results thus demonstrated that within a given plasmid PTU, the conjugation parameters may exhibit significant variation.

## Discussion

Understanding the factors that facilitate or curtail plasmid conjugation is essential to devise strategies against the propagation of antibiotic resistances. Plasmid propagation is a complex process, in which transmission, vegetative growth and plasmid-host co-adaptation play important roles [49]. Plasmid transmission is classically modelled as a mass-action kinetics process, but our results showed that this assumption breaks at moderate cellular densities of less than 0.1 cells per square micron. From this density onward, conjugation resembles an FDT process, where the number of T cells produced per donor per unit time remains constant. By applying a model based on Holling´s type II functional response [50], we were able to build a quantitative framework that faithfully recaptures the DDT/FDT regime of plasmid conjugation. Our model depends on two parameters: the encounter rate ($k_{on}$, with units corresponding to area per time, µm2·h⁻¹) and the engagement time, $\tau$ (with units of time, h). The interplay between these two parameters (Eqs.2 and 3) determines the cell density at which

density or frequency become limiting. By fitting conjugation frequencies obtained at different total cell densities we were able to obtain estimates of both parameters for plasmids from the PTU-P1, PTU-N1 and PTU-W. Estimates of the encounter rate, $k_{on}$, were in all cases in the range of hundreds of square microns per hour, which roughly correspond to a searching radius of 6–10 $\mu m^2$ per hour. These results are in accordance with estimates and recent direct observations of the effective distance for plasmid transmission, obtained for PTU-$F_E$ and PTU-P1 plasmids [51–55].

Interestingly, although both the encounter rate and the engagement time showed significant variation between plasmids, the cell density at which plasmids transitioned from one regime to the other was similar in all cases. Transition from a density-dependent to a frequency-dependent regime occurred at cell densities of approximately 0.02–0.05 cells per $\mu m^2$ (that is, a 1 $\mu m$-long cell every 20–50 $\mu m^2$). This figure roughly corresponds to a cellular density where each potential donor would be, at most, contacting 1 recipient. This is a surprisingly low cellular concentration, indicating that in denser environments such as biofilms [56–58], soil [59] or the gut microbiota [60] conjugation is frequency-determined. This indicates that in a significant proportion of microbial environments plasmid conjugation is limited by the engagement time, rather than the probability of encounters, as assumed by classical models of plasmid transmission [3,4,9].

This frequency limitation explains many experimental results obtained for conjugation on solid surfaces and biofilms. In these experiments, a population of donors encountering one of recipients is typically able to generate only a "border" of transconjugants [18,61–64]. As engagement times of conjugative plasmids are on the same order of magnitude as the doubling times of donors and recipients [44], the layer of contact of recipients is pushed backwards at the same speed that the layer of transconjugants is created. When frequency becomes limiting, the factor constraining plasmid propagation is the time until the exhaustion of resources, compared to the engagement time τ. The higher the number of effective transfers achievable by the plasmid before resources run out, the higher the conjugation rate. This explains why most plasmid propagation experiments on highly dense environments produced similar low invasion levels, except those continuously refreshing the growth medium [18,65]. Thus, to accurately measure the conjugative efficiency, experiments with short mating times (30–90') and an excess of recipients (1:100 D:R ratio) are likely to yield more precise estimations. Results obtained in longer experiments, and with 1:1 D to R ratios will be confounded by vegetative growth and the lower probability of D-R encounters, respectively.

The engagement time, τ, was found the most significant constraint for transfer. The factors that influence this parameter, however, are unclear. The fact that plasmids and their cognate transfer machineries exhibit slightly different engagement times (0.5 h for pKM101 and 0.33 h for $Tra_N$) suggests that factors outside the conjugation region may play a role. The difference in size of the DNA being transferred is the most obvious candidate (pKM101 is 35 kb and the vector mobilized by $Tra_N$ is approximately 5 kb). However classical experiments using HFR strains demonstrated that DNA transfer is fast, and the entire chromosome can be readily conjugated in approximately 100 minutes [66,67]. Other experiments also demonstrated that plasmid-encoded fluorescent proteins are expressed in transconjugants after 15 minutes of mating, indicating that the whole process of transfer takes shorter times [68,69]. More recently, the timescale of the whole process of plasmid mobilization has been experimentally measured and seems to be in accordance with previous estimations [70]. Altogether, these results suggest that the engagement time is not merely the time required to pump the DNA into the recipient. Instead, other factors conditioning the establishment and resolution of an effective conjugative pair may be at play. Plasmid copy number may be another possible contributor to τ. R388 and pKM101 plasmids are relatively low copied,

compared to the vectors mobilized in assays using $Tra_W$ and $Tra_N$ mediated mobilizations. Moreover, the different IncW plasmids assayed in Fig 4 also present different copy numbers [45], which could contribute to the differences observed in transfer rates. Further experiments systematically analyzing the impact of plasmid copy number on the transfer rate are required to clarify this end.

The searching rate $K_{on}$, establishes how easily a successful D-R pair is established upon encounter. Physical factors involving cell to cell adhesion are likely to contribute to this rate. It is known that conjugation is favored when other, unrelated conjugative plasmids express their pili in the same donor, or even the recipient cell [32]. This suggests that increasing adhesion may facilitate transfer, a phenomenon that was also observed introducing synthetic adhesins in D and R cells [71]. Plasmids pKM101 and R388 showed a significant difference in their $K_{on}$ (Table 1), a phenomenon which may be related to differences in the ability of the plasmid to facilitate cell to cell adhesion. Plasmid pKM101 contains *pep*, an adhesin-like protein that is absent in R388 and that has been shown to facilitate conjugation [72]. Also, the only protein that slightly increased the conjugation efficiency in R388 when overexpressed was the VirB5 adhesin TrwJ (Fig 3). Similar results have been obtained in pKM101 [73], pTi [41], and R64 [74], thus reinforcing the idea that increasing adhesion improves the conjugation rate. In fact, recent research shows that IncI and IncF plasmids selectively "pick" their Recipients via pilV or TraN-OMP surface interaction respectively [37,38,40,75].

The fact that plasmids from the same PTU exhibited different encounter rates and engagement times was surprising. It indicates that conjugation efficiency is not a conserved feature among the members of a PTU, despite the overall sequence conservation (S6 Fig). In the case of W plasmids, sequence divergence concentrates in the *mob* and replication regions, and are the highest between R388 and R7K plasmids [76,77], and changes in the TrwJ adhesin are also observed among PTU-W plasmids (S6 Fig).Thus, the molecular reasons behind the differences in encounter rates and searching times exhibited by members of this PTU cannot be pinpointed. Interestingly, R7K showed the lowest encounter rate and the longest engagement time, which resulted in the slowest invasion dynamics (Fig 4). From all PTU-W plasmids, R7K is the only one that was originally isolated from a host outside Enterobacteria. Its poor performance may represent a maladaptation to *E. coli*, compared to the rest of PTU-W plasmids. Previous analyses have shown that plasmid-host co-evolution has a major impact on key plasmid properties such as host range and burden [49,78–80] It is thus possible that plasmid transmission also depends on co-adaptations between the plasmid and the host.

Our results indicate that a single rate is not enough to parametrize the conjugation rate of a plasmid in a particular host, as it presents a complex relationship towards cell density and time. By systematically measuring these parameters in plasmids at different stages of co-adaptation, we may be able to correlate transfer efficiencies with genes and mutations. Our model thus provides an analytic framework to unravel the mechanisms and constraints that regulate plasmid propagation in bacterial populations.

## Materials and methods

Plasmids and strains employed in this work are detailed in S1 and S2 Tables.

**Cloning.** Plasmid constructs used for gene dosage experiments were built using the Gibson assembly method [81], using pBAD33 PCR products as vector and genes and operons from *tra* region of R388 as insert (to be inserted downstream the arabinose promoter of pBAD33 [82]). All fragments were PCR amplified with Phusion DNA polymerase (Thermo Fischer),

gel-purified and digested with DpnI restriction endonucleases (Promega) before assembly. *E. coli* DH5α strain was used for electroporation and selective culture purposes. Gibson assembly products were introduced in *E. coli* by electroporation. To build the mobilizable plasmid pSEVA321::mobW, the region containing *oriT* of RP4 included in vector pSEVA321 (coordinates 1278 to 1586 of GenBank Acc. No. JX560322.1) was replaced by the *oriT-trwABC* region of R388 (coordinates 11153-16299 of GenBank Acc. no. BR000038) using Gibson assembly. Maps of the plasmids containing the *mob* regions of W and N plasmids are shown in S7 Fig.

**Tra$_W$ and Tra$_N$ constructions.** The MDS42::*tra$_W$* strain was constructed by cloning the *mpf* (also called *tra* operon) region of R388, comprising all genes (*trwD-korB*) required to build up the conjugative pili and inserting this whole region in a MDS42 *E.coli* strain [83] using Wanner-Datsenko protocol [84]. The *lacZYA* genes of *E. coli* strain MDS42 (coordinates 287438-292616 of GenBank Acc. No. NC_020518.1) were used as target for the chromosomal insertion. The *trwD-korB* region of R388 (coordinates 1-11226 of GenBank Acc. No. BR000038.1) was PCR-amplified in three overlapping fragments. The kanamycin-resistance gene (Km$^R$) surrounded by FLP recombinase recognition sites (FRT) was amplified from pKD4 (AY048743.1). Finally, 1kb fragments, homologous to the *lacI* and *cynX* chromosomal genes, were amplified from MDS42 genomic DNA. The six fragments were assembled (*lacI-trwD-korB-KmR-cynX*) and introduced by electroporation into MDS42 competent cells expressing the Lambda-Red site-specific recombinase [84]. Km$^R$ colonies were checked by PCR to confirm the Tra$_W$ integration in the expected location, and the genome of a selected colony was completely sequenced by Illumina. The phenotype of the MDS42::*tra$_W$* strains was checked by culturing in a chromogenic medium (agar Brilliance) that differentiates lac- and lac+ phenotypes. The Km$^R$ cassette was subsequently deleted following the procedure described by Datsenko and Wanner [84]. Resistant derivatives of MDS42::*tra$_W$* were obtained by plating saturated cultures of this strain in LB-agar plates supplemented with Rif10, Nx20 or Sm300.

MDS42::Tra$_N$ strain was constructed by amplifying and Gibson-assembling the *mpf* region of pKM101 (genes *korB-traG*) in a pOSIP vector [85], then electroporated in a DH5α *pir$^+$* strain before PCR comprobation, miniprep purification and introduction by electroporation in a *pir$^-$* MG1655 strain -following St-Pierre et al. protocol- where it recombined at *attC* position in the *E. coli* chromosome [85].

**Conjugation and mobilization experiments.** Unless otherwise stated, mating assays were performed using BW27783 [86,87] Nal$^R$ (donor) and Rif$^R$ (recipient) cells that had been grown to saturation overnight, at 37 °C, in 10 ml LB media supplemented with appropriate antibiotics. Antibiotic concentrations used were nalidixic acid 20 µg/ml, rifampicin 10 µg/ml, kanamycin 50 µg/ml, chloramphenicol 25 µg/ml, trimethoprim 20 µg/ml and ampicillin 100 µg/ml. Cells were washed in equivalent volumes of fresh LB without antibiotics and allowed to grow at 37 °C for 3h more. These cultures were collected, centrifuged at 4000g for 10' and concentrated in 1 ml of fresh LB, then mixed at desired D:R ratios. Conjugation on solid surfaces were performed in Corning Costar 24 well plates to which 1 ml of LB-Agar had been added at least 3 days before, to solidify and dry. 15 µl of mating mix were placed on top of the LB-Agar and let to conjugate for 1 h. at 37 °C, except when indicated otherwise. Cells were then resuspended using 1 ml of sterile PBS, subject to serial dilution and plated on LB-agar with appropriate antibiotics.

Conjugation assays using serial dilutions to discern between FDT and DDT were performed as indicated above with the following modifications: For conjugation on solid surfaces, cell cultures of donor (10 ml) and recipient (100 ml) cells were grown to saturation on LB broth containing the appropriate antibiotics. Cells were washed, refreshed for 3h in LB without antibiotics, concentrated ~100x in fresh LB and then mixed in 1:100 Donor to

Recipient ratio before being serially diluted in LB (dilution ranging from $10^2$x to $10^{-4}$x). From each dilution, 100 μl of cells were deposited on top of a 10 cm diameter LB-agar plate (also prepared at least 3 days before to ensure proper LB agar drying) and incubated for 1h at 37 ºC. Cells were resuspended in 1000 μl of PBS, serially diluted and appropriate dilutions were plated on LB-agar plates with antibiotics to count the number of D, R and T cells present. Every D, R and T count corresponded to three technical replicates. Cell densities (in cells/μm²) were calculated dividing the average of these 3 counts per total surface: 2 cm² in 24-well plates, 58 cm² in Petri plates.

The same protocol was applied in mating experiments at different times, with only two changes: a 1:10.000 ratio was also used (by putting 1/100 less donor volume in the same amount of 10x concentrated recipient cells, as usual) and PBS was used to abort conjugation at different times: 30, 60, 90, 120, 150 and 180 minutes. Several methods for measuring this efficiency have been used [16,32,88–92] and discussed elsewhere [3,10,19,26,27], but we will use T/D as a measurement of conjugation (or mobilization) efficiency on surfaces because at very short times (<3 h) it is free from factors like differences in growth or population structure, still depicting very intuitively the differences in transmission efficiency, and its underlying causes. The cell density used at the beginning of these experiments was approximately 1 cell/μm².

Experiments measuring gene dosage effect on conjugation were conducted as the rest of conjugation experiments explained above, except for the donor and recipients were grown overnight on LB with antibiotics and glucose 0.5% (for pBAD33 arabinose promoter repression) and washed and allowed to grow for 3 h more in LB without antibiotics with glucose 0.5% or arabinose 50 μM (for arabinose promoter induction). Each experiment was conducted simultaneously with a control donor containing the conjugative plasmid and pBAD33 with no cargo. No transfer of pBAD33-derived plasmids was observed. Conjugation efficiencies were determined as T/D and ratios mean conjugation efficiency (three replicates) for the plasmid + pBAD33::trwX donors divided by mean conjugation efficiency (three replicates) for the plasmid + pBAD33 control were represented in graphs. In this case, if no change in conjugation efficiency is observed, the expected mean ratio clone:control should be 1, while if efficiency is improved by supplementing a gene or operon, the ratio should be greater than 1. In this case, DH5α strain was used as donor to avoid recombination between vector cargo and R388wt.

Conjugation experiments in buffered LB were performed at a pH = 7.4 using $Na_2HPO_4$ 10 mM and $NaH_2PO_4$ 1.8 mM and pH adjusted with HCl 35% and NaOH 1M. Conjugation experiments using filtered LB agar were performed as described, but using LB medium filtered with a 0.22 μm filter membrane instead of using high-pressure sterilized LB. On the same manner, TB agar experiments were performed as usual, but using autoclaved Terrific Broth instead of LB, with the same medium to agar ratio.

Statistical tests. In order to check if conjugation efficiency measurements (as T/D cell counts) did follow a normal distribution we performed a Shapiro-Wilk and an Anderson-Darling test on two independent arrays of 16 measurements of R388 conjugation efficiency each. Both groups of T/D counts showed no normality, but when the analysis was performed on their log-transformed values they showed normal distribution (Shapiro-Wilk, W = 0.97 both groups, Anderson-Darling test $A^{*2}$ = 0.26 and 0.21 respectively), thus indicating log-normal distribution, as suggested before [32,93].

Paired comparisons of log(T/D) values were performed using two-sided t Student's test, while multiple comparisons were performed using ANOVA. Unless otherwise specified, a significance level of α = 0.05 was taken as good enough to reject null hypothesis.

Nonlinear fitting of conjugation efficiencies at different cell concentrations was performed with nonlinear curve fitting package of GraphPad Prism v8.0.2 using custom equations (namely, Eqs 2 and 3).

## Supporting information

**S1 Table. Strains used in this work.**
(XLSX)

**S2 Table. Plasmids used in this work.**
(XLSX)

**S1 Fig. A) Growth of Donor (solid circles) and Recipients + Transconjugants (hollow circles) during the first hour of mating in conjugation experiments of R388 (black, right) and pKM101 (red, left) carrying cells at 1:10,000 D:R ratios as detailed in** Fig 2A **in the main text.** ANOVA testing comparing the four populations at 0 and 60 minutes after mating began gave p > 0.05 in all but R388 Recipients' growth. B) Growth curves for donor and recipients in matings using plasmid R388 (black circles, upper graph) and pKM101 (red circles, lower graph) fitting showed that doubling times were >60 minutes for all strains analyzed.
(TIFF)

**S2 Fig. DDT/ FDT transition in conjugations with plasmid pOX38 and R64 on solid media.** Transconjugants per donor ([T]/[D], y axis) measured at different Recipient densities ([R] ≈ [N], x axis) in E. coli BW27783 mating on solid LB-agar surfaces, 1 Donor to 100 Recipients and 1 h. conjugation time for plasmids pOX38 (IncFI) and R64 (IncI1). As in Fig 1 in the main section, the black line represents the ideal DDT regime, where different $k_{on}$ produce different y-intercepts. The red line represents the FDT regime, where the conjugation efficiency is just $1/\tau$ and the dotted curve corresponds to the fitting to Eq.3. Every black dot corresponds to the average of 3 technical replicates.
(TIFF)

**S3 Fig. DDT/ FDT transition in conjugations with plasmid R64 in liquid media.** Transconjugants per donor ([T]/[D], y axis) measured at different Recipient densities ([R] ≈ [N], x axis) in E. coli BW27783 mating in liquid LB, 1 Donor to 100 Recipients and 1 h. conjugation. Every dot corresponds to the average of 3 technical replicates.
(TIFF)

**S4 Fig. A)** Results from stochastic simulations showing the conjugation frequency (T/D, y axis) against time (min, x axis) obtained at different cellular densities (D). Shadowed areas correspond to 1 and 2 generation times. **B)** Comparison between computational simulations (dots) and theoretical predictions (lines) for the conjugation frequency (x axis, T/D) obtained at different cellular densities (x axis), at different generation times (legend). The normalized estimation error obtained at different generation times is shown in the inner graph. The normalized estimation error was measured as the difference between the conjugation frequency computed and estimated, divided by the average of the two. **C)** Effect of growth differences between donor and transconjugant cells on the estimation of the conjugation frequency. Computed (dots) and predicted (line) conjugation frequencies (x axis, T/D) obtained at different cell concentrations (x axis), when the transconjugants experience a growth deficit with respect to the donor cells indicated in the legend. As shown in the figure, a significant deviation from the estimated values was observed only for growth deficits >50%.
(TIFF)

**S5 Fig. A)** Doubling times in LB broth for recipient strains (*E. coli* Bw27783) and donor cells (Bw27783 containing the plasmids indicated in the legend) for the model plasmids employed in this work. **B)** Doubling times in LB broth for recipient strains (*E. coli* Bw27783) and

donor cells (Bw27783 containing the plasmids indicated in the legend) for the plasmids from PTU-W employed in this work.
(TIFF)

**S6 Fig. A)** Sequence conservation among PTU-W plasmids. Alignment of sequences indicating the synteny on Replication genes (indicated in blue), STB (brown), MOB (light blue) and MPF (green) operons in the 4 plasmids used in Fig 4. Antimicrobial resistance genes are indicated in red, transposases in orange. **B)** Conservation of the adhesin among PTU-W plasmids. TrwH-TrwG(maroon)-TrwJ(light red) complex from R388 pilus tip, as taken from PDB (https://www.rcsb.org/structure/8RT9). Mutated residues in pIE321, R7K or both are indicated in green, blue and pink, respectively.
(TIFF)

**S7 Fig. Scheme of *mobW* and *mobN* plasmids used in this study, carrying the MOB operon of R388 and pKM101 respectively.** Antimicrobial resistance gene markers are coloured in red, mobilization genes are coloured in green, oriT and oriV sequences are indicated in blue.
(TIFF)

**S1 Calculations. A detailed set of derivations leading to analytical solutions for: (1) plasmid mobilization, (2) plasmid conjugation using the Holling's Type II model, (3) the contribution of vegetative growth to the conjugation frequency, and (4) testing the approximations using chemical master equations.**
(PDF)

## Author contributions

**Conceptualization:** Jorge Rodriguez-Grande, M. Pilar Garcillán-Barcia, Fernando de la Cruz, Raul Fernandez-Lopez.

**Data curation:** Jorge Rodriguez-Grande, Yelina Ortiz, Daniel Garcia-Lopez, M. Pilar Garcillán-Barcia, Raul Fernandez-Lopez.

**Formal analysis:** Raul Fernandez-Lopez.

**Funding acquisition:** Raul Fernandez-Lopez.

**Investigation:** Jorge Rodriguez-Grande, Yelina Ortiz, Daniel Garcia-Lopez, M. Pilar Garcillán-Barcia.

**Methodology:** Jorge Rodriguez-Grande, Yelina Ortiz, M. Pilar Garcillán-Barcia, Raul Fernandez-Lopez.

**Project administration:** Raul Fernandez-Lopez.

**Resources:** Raul Fernandez-Lopez.

**Supervision:** Raul Fernandez-Lopez.

**Writing – original draft:** Jorge Rodriguez-Grande, Raul Fernandez-Lopez.

**Writing – review & editing:** Jorge Rodriguez-Grande, Yelina Ortiz, Daniel Garcia-Lopez, M. Pilar Garcillán-Barcia, Fernando de la Cruz, Raul Fernandez-Lopez.

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
