## [Decision Letter · Decision Letter 0]

6 Oct 2024

Dear Dr Fernandez,

Thank you very much for submitting your Research Article entitled 'Fundamental parameters governing the transmission of conjugative plasmids' to PLOS Genetics. And, many apologies for the late feedback on your manuscript.

The manuscript was fully evaluated at the editorial level and by independent peer reviewers. As you will see, reviewer #1 only has minor comments, which can all be addressed by changes to the text. Reviewer #2 has more significant concerns, some of which will have to be addressed experimentally.  Based on the reviews, we will not be able to accept this version of the manuscript, but we would be willing to review a much-revised version. We cannot, of course, promise publication at that time.

If you decide to revise the manuscript for further consideration at PLOS Genetics, please aim to resubmit within the next 60 days, unless it will take extra time to address the concerns of the reviewers, in which case we would appreciate an expected resubmission date by email to plosgenetics@plos.org.

If present, accompanying reviewer attachments are included with this email; please notify the journal office if any appear to be missing. They will also be available for download from the link below. You can use this link to log into the system when you are ready to submit a revised version, having first consulted our Submission Checklist .

PLOS has incorporated Similarity Check , powered by iThenticate, into its journal-wide submission system in order to screen submitted content for originality before publication. Each PLOS journal undertakes screening on a proportion of submitted articles. You will be contacted if needed following the screening process.

To resubmit, log into your Editorial Manager account and select the option 'Revise Submission' in the 'Submissions Needing Revision' folder.

We are sorry that we cannot be more positive about your manuscript at this stage. Please do not hesitate to contact us if you have any concerns or questions.

Yours sincerely,

Lotte Søgaard-Andersen, Ph.D., M.D.

Section Editor

PLOS Genetics

Reviewer's Responses to Questions

**Comments to the Authors:**

Reviewer #1: I have to declare that I reviewed a previous version of this manuscript for PLoS Biology. In general, I found the work robust, relevant and interesting. Below, I include my previous comments and suggestions, after removing those that were already corrected by the authors (most of them).

In this manuscript, Rodriguez-Grande and collaborators dissect the conjugations dynamics of plasmids in order to better inform the parametrization of the conjugations process in bacterial population dynamics models. Conjugative plasmids are arguably the main vehicle for the dissemination of antimicrobial resistance (AMR) genes. Given the great threat that AMR presents to public health worldwide, understanding the dissemination of AMR plasmids in bacterial communities is of paramount importance. Most mathematical models assume the principle of mass action to parametrize conjugation, which assumes that the probability of two cells encountering each other and engaging in conjugation is proportional to the densities of both the plasmid-free and plasmid-bearing subpopulations. In this work, the authors show that conjugation is a density-dependent process only at low population density, but it becomes frequency dependent at higher bacterial concentrations. The authors convincingly

describe these dynamics and the key parameters that determine them, and they move on to offer the mechanistic insights underlying them. In general, I have really enjoyed reading this manuscript. I think it provides key new results that will be crucial to perform better and more effective models to predict the dissemination of AMR in bacterial populations. Below, I provide a few minor comments and suggestions:

The title could be somehow more informative and hopefully attract more readers towards the paper, as it reads now is a bit vague.

It would be useful to have a supplementary figure with the schematic representation of MOBN and MOBW and also to indicate which plasmids are these in Table 3.

Maybe a supplementary figure with an alignment of plasmids R388, pMBUI4, pIE321 and R7K could help. The species of isolation could also be indicated there.

Alvaro San Millan

Reviewer #2: Rodriguez-Grande et al have considered how the conjugation dynamics of various plasmids depends on plasmid characteristics (transfer genes) and environmental parameters (cell density, temperature, medium composition). In my opinion, the most interesting and novel aspect is the Holling type II response. However, these findings are only loosely connected to the later parts of the manuscript. I also have a number of methodological concerns about the assays performed to quantify conjugation, some of which may affect the interpretation of the presented results.

1.) Did the authors perform on-plate mating controls for any of the experiments, most importantly for Fig 1? Varying the density of cells may well affect the amount of unwanted mating during the selection plating part of the assay (https://link.springer.com/article/10.1007/s00248-024-02427-7 ; https://doi.org/10.1016/j.plasmid.2023.102685 ). This could lead to biased conjugation rate estimates at higher cell densities. In a related vein, did the authors check the sensitivity and specificity of their selection assay? Is it the same for all considered plasmids?

2.) The introduction to the first results section is misleading. “In a pure FDT, however, the number of T/D cells obtained remains constant regardless of D and R concentrations (Begon et al., 2002).”

I do not think this is generally true. The per capita contact rate increases linearly with density (DDT) or is flat (FDT) - but it is too much to say that the ratio T/D is flat. For instance if the T and D cells have a different growth rate, even in absence of conjugation (whether following DDT or FDT) the ratio T/D would vary over time.

In general, I think the authors should spell out more early and clearly that they use T/D as an estimate of a conjugation rate, and under which conditions this assumption is warranted.

2’) For the short mating assays in Fig 1 T/D may be fine, but for the 3h assays (shown in Figs 2/3/4) Supplementary Fig 1 suggests that the cells are actually growing, possibly at a different rate across different conditions. To exclude that different growth costs of the different constructs are driving these dynamics, it would be advisable to determine the growth rates of the various mutants.

3.) Is it possible to connect the searching rate and handling times to more established metrics for the conjugation rate? It seems the searching rate would be proportional to a bulk conjugation rate on plates?

4.) Figure 2 shows common conjugation assays, and the relationship to the previous discussion on the Holling Type II response. To assess the effect of the searching and handling time, it would have been nice to compare the dynamics at low and high cell density. What density were these assays performed at?

Fig 1: How were the #cells per um determined? The methods section does not seem to suggest that initial time point measurements were taken?

Also, please include error bars from the 3 replicates (either in the main text or a supplementary figure).

Fig. 2:

a) it is confusing that statistical tests were performed for only some but not all comparisons. Were the others not significant? Why did the authors choose not to report tests in panels B and D?

b) Which cell density was used for the time course measurements? Where do the observed T/D measurements fall on the Holling Type II response curve? How would the measurements differ if taken at a different point on the curve?

Fig 4: The panels are wrongly referenced in the accompanying text (4A as 4B and vice versa).

What is the confidence interval around the fit of searching rate and handling time for these plasmids?

Other points:

- Please add linenumbers to your future submissions.

- Fig 1AB: is it intended that both processes lead to the same number of offspring in the same size population? I struggle to understand the message in these figures.

- Suggestion: if the sequences of the PTU-W plasmids are available, it would be interesting to see whether R7K has a TrwJ mutation.

**Have all data underlying the figures and results presented in the manuscript been provided?**

Reviewer #1: Yes

Reviewer #2: **No: ** The data underlying the various figures is not available. The data availability statement reads "available upon request", which did not allow this reviewer to assess them.

PLOS authors have the option to publish the peer review history of their article (what does this mean? ). If published, this will include your full peer review and any attached files.

**Do you want your identity to be public for this peer review?** For information about this choice, including consent withdrawal, please see our Privacy Policy .

Reviewer #1: **Yes: ** Alvaro San Millan

Reviewer #2: No

---

## [Editor Report · Decision Letter 1]

31 Dec 2024

Dear Dr Fernandez,

We are pleased to inform you that your manuscript entitled "Encounter rates and engagement times limit the transmission of conjugative plasmids" has been editorially accepted for publication in PLOS Genetics. Congratulations!

Yours sincerely,

Lotte Søgaard-Andersen, Ph.D., M.D.

Section Editor

PLOS Genetics

Lotte Søgaard-Andersen

Section Editor

PLOS Genetics

Aimée Dudley

Editor-in-Chief

PLOS Genetics

Anne Goriely

Editor-in-Chief

PLOS Genetics

Comments from the reviewers (if applicable):

**Data Deposition**

http://datadryad.org/submit?journalID=pgenetics&manu=PGENETICS-D-24-00879R1

**Press Queries**

---

## [Editor Report · Acceptance letter]

PGENETICS-D-24-00879R1

Encounter rates and engagement times limit the transmission of conjugative plasmids

Dear Dr Fernandez-Lopez,

We are pleased to inform you that your manuscript entitled "Encounter rates and engagement times limit the transmission of conjugative plasmids" has been formally accepted for publication in PLOS Genetics! Your manuscript is now with our production department and you will be notified of the publication date in due course.

With kind regards,

Anita Estes

PLOS Genetics

On behalf of:
